# Scalable Multimodal Fine-tuning for Foundation Models via Mixture-of-LoRA

## Abstract

Adapting pre-trained Large Language Models (LLMs) for multimodal tasks presents a significant challenge, often hindered by the prohibitive computational cost of full fine-tuning. In this work, we introduce Mixture-of-LoRA (MoL), a novel and parameter-efficient fine-tuning framework that enables LLMs to seamlessly process and integrate multimodal inputs. MoL combines the efficiency of Low-Rank Adaptation (LoRA) with the modality-specialized design of Mixture-of-Transformers (MoT). Our approach injects small, trainable, modality-specific LoRA adapters into the frozen layers of a pre-trained LLM. While each modality's tokens are processed by these dedicated adapters to learn specialized features, the global self-attention mechanism remains intact, allowing for rich cross-modal fusion within the original LLM architecture. This design efficiently adapts the model to understand diverse data types–such as text, images, and speech–while retaining and leveraging the vast knowledge of the foundational model. Through extensive experiments, we demonstrate that MoL effectively enables pretrained foundation models to *understand* and *generate* multimodal tokens. Our work provides an effective and scalable solution for building multimodal systems from existing unimodal foundation models.

## 1 Introduction

Large language models (LLMs) (Touvron et al., 2023; Abdin et al., 2024; Yang et al., 2024a; OpenAI et al., 2024; Grattafiori et al., 2024; DeepSeek-AI et al., 2025) have received increasing attention from both researchers and practitioners due to their capabilities that have expanded beyond text modality. For instance, since the rise of decoder-only models originally proposed for natural language (Radford et al., 2019), many models can now process diverse modalities, e.g, text, audio, image, or videos.

World models have now extended to take multimodal inputs such as image and text modalities, (Aghajanyan et al., 2022a; Liu et al., 2023b; Team, 2024), speech and text (Fathullah et al., 2024; Yu et al., 2024; Chu et al., 2024), or video and text (Jiang et al., 2025; Ye et al., 2025). Some approaches have even extended the ability of models to take more than two modalities at a time. For instance, Liang et al. (2025) train a model from scratch to take audio, image, and text as inputs. Similarly, Lyu et al. (2023) put forward a model able to receive inputs in the form of audio, image, video, and text.

Two approaches to make LLMs multimodal have generally been considered. A first approach consists of taking a pretrained text-only LLM as a base model and fine-tuning it to take multimodal inputs. This usually involves including an adapter module that maps the other modalities' tokens to the representation space of the LLM. These approaches are particularly interesting because they leverage the vast amount of learned information already present from their extensive text-based training. Also, those approaches are cost-efficient as they avoid retraining a model from scratch, which has been known to induce significant computational costs (Liang et al., 2025). However, this line of approach usually requires carefully curating both the modalities' feature extractors and mapping modules. This often involves producing different mapping modules for each modality, thus making hyperparameter optimization more complex. Some recent work (Laurençon et al., 2024) investigated the impact on performance of model architectures, in particular connector modules, in the context of Vision-Language Models. Similarly, Verdini et al. (2025) demonstrates that the ideal adapter and feature extractor depend on the target task for speech-text models. Additionally, previous work (Das et al., 2024; Thimonier et al., 2025) highlights the sensitivity of these approaches to the training

curricula. These findings highlight the complexity of fusing new modalities into pretrained LLMs. Finally, multimodal fine-tuning is usually restricted to enabling the base LLM to *understand* new modalities and rarely involves teaching the model to *generate* other modalities than text.

The second type of approaches includes training foundational models from scratch to take multimodal inputs. While these approaches perform best on a wide range of multimodal tasks, they require a significant computational cost to train. For instance, models like Chameleon (Team, 2024) require 1.5 trillion text-image tokens, 2.9 trillion text-only tokens, and 400 billion interleaved tokens.

Recently, Liang et al. (2025) have proposed a novel architecture to train multimodal foundational models: Mixture-of-Transformers (MoT). Their proposed approach is motivated by the finding that multimodal foundational models display clustering by modality across layers. In short, their approach disentangles the different modalities in the token sequence within each attention layer. All modalities are processed independently to produce the query, key, value, and output matrices. Notably, the obtained modality-specific representations are concatenated (in their original order), and self-attention is applied to the concatenated representations. We extensively discuss this approach in section 3.2. While significantly more efficient than existing methods, as the required flops for the same performance decrease, this approach still requires retraining a model from scratch with trillions of multimodal tokens.

We build on this approach and propose a novel multimodal fine-tuning approach of LLMs, **Mixture-of-LoRA (MoL)**, that leverages per-modality LoRA adapters (Hu et al., 2022). In our proposed approach, we freeze the weights of a pretrained text-only LLM, in particular the weights of the query, key, value, and output matrices, and add on top of them, per-modality LoRA adapters (see Fig. 1). Instead of learning an entire weight matrix per modality at each attention head and multi-attention layer, we rely on efficient fine-tuning by considering low-rank matrices that significantly reduce the overall training cost.

We evaluate our fine-tuning approach focusing on relatively small LLMs ($\leq$ 3B parameters) and restricting our experiments to the pretraining stage of this multimodal setting. We experiment on two settings:

1. **Autoregressive objectives for text and images**, coined Chameleon setting by Liang et al. (2025). We observe that our approach efficiently enables the model to both understand and generate image tokens. We compare to a baseline LoRA approach and demonstrate the superiority of our approach to this vanilla case.

2. **Three-modality setting (Text+Image+Audio)**. We demonstrate the capacity of our fine-tuning method on a three-modality setup by adapting a text-only LLM to the same task but involving text, image, and speech. Our experiments demonstrate that it can successfully understand and generate all three modalities.

## 2 RELATED WORKS

**Multimodal LLMs** While the first LLMs solely focused on natural languages, a large spectrum of multimodal foundation models has been proposed in the literature. Multimodality in foundation models first involved multimodal *understanding* and relied on modality-specific feature-extractor and mapping modules. Traditionally, images are encoded in the LLMs' representation space using carefully curated mapping modules, using late-fusion techniques (Alayrac et al., 2022; Chen et al., 2022; Liu et al., 2023b). Some recent work (Vallaeys et al., 2025) has investigated the optimal adapter to map the audio/image feature representation to the LLM representation space. As discussed in section 1, these approaches are often less costly to train than fully-trained multimodal models as they rely on pre-trained LLM backbones and require fewer tokens to fine-tune.

The other line of approaches that require retraining from scratch a foundational model, e.g., MoT Liang et al. (2025), Chameleon (Team, 2024), Unified-IO (Lu et al., 2022), CM3 (Aghajanyan et al., 2022b), or CM3Leon (Yu et al., 2023), enable both visual *understanding* and *generation*. To that end, modalities like images or audio need to be tokenized using a discrete dictionary using pre-trained models that involve fixed codebooks (van den Oord et al., 2017; Razavi et al., 2019; Esser et al., 2020; Liu et al., 2023a). These approaches thus allow auto-regressive generation of other modalities than just text, as the LLM's dictionary can be extended to include these new modality-specific tokens.

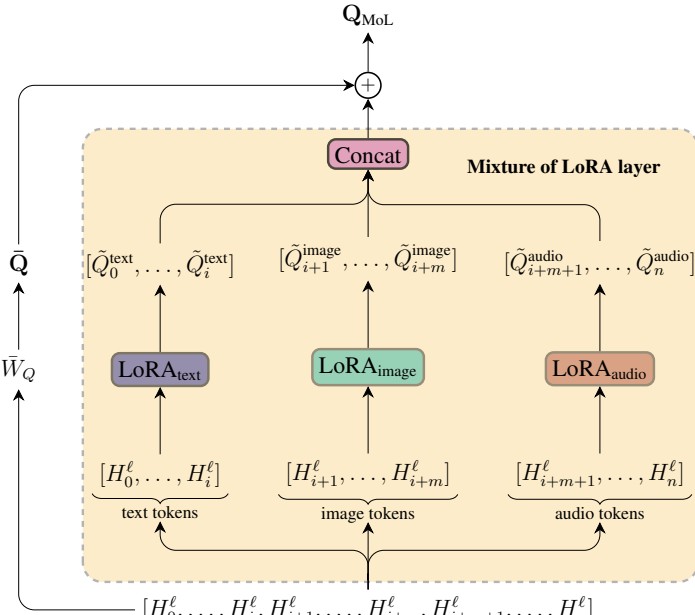

Figure 1: **Mixture of LoRA Layer**. Mixture of LoRA (MoL) layer for the query generation of an attention head at layer $\ell$. For simplicity, we omit the head indices. The weights of the pretrained LLM for the corresponding layer, $\bar{W}_Q$, are frozen and produce the query matrix $\bar{\mathbf{Q}}$. The MoL layer first disentangles the tokens corresponding to each modality $m \in \mathcal{M}$ and processes them independently using modality-specific LoRA adapters. The obtained query matrices, $\{\tilde{\mathbf{Q}}^m\}_{m \in \mathcal{M}}$, are then concatenated in their original order to match the original query matrix $\mathbf{Q}$ dimension. As in the standard LoRA setting (Hu et al., 2022), $\tilde{\mathbf{Q}}$ is scaled by $\frac{\alpha}{r}$ and added to $\bar{\mathbf{Q}}$ to obtain the final query representation of the token sequence $\mathbf{Q}_{\text{MoL}}$.

**LLM fine-tuning**    A prevalent method for adapting foundation models is fine-tuning. This process specializes a general-purpose model by continuing its training on a comparatively small, task-specific dataset, enabling its application to specific domains of interest (Devlin et al., 2019). Initial approaches to parameter-efficient fine-tuning (PEFT) inserted learnable adapter modules between a model's frozen layers (Rebuffi et al., 2017; Houlsby et al., 2019; Lin et al., 2020). The influential LoRA framework (Hu et al., 2022) advanced this by instead decomposing the weight update matrix of a layer into two trainable low-rank matrices, which are learned in parallel to the frozen original weights. While some work leverages LoRA adapters to fine-tune pretrained LLMs to *understand* new modalities like audio (Das et al., 2024) or images Liu et al. (2023b), the present work is the first to leverage LoRA by using per-modality adapters during training.

## 3    METHOD

In the present section, we briefly overview the vanilla attention (Vaswani et al., 2017) mechanism, then we present the mechanisms underlying the MoT architecture (Liang et al., 2025), and then discuss MoL in light of this.

### 3.1    VANILLA ATTENTION

Let $\mathbf{X} = (\mathbf{x}_1, \ldots, \mathbf{x}_n)$ be the input token sequence, where $\mathbf{x}_i$ belongs to a modality $m_i$ where $m_i \in \mathcal{M} = \{text, speech, image\}$. A typical transformer layer consists of the following,

$$
\begin{aligned}
a &= \text{Attn}\left(\mathbf{X}, \theta_{\text{attn}}\right) \\
\mathbf{h} &= \mathbf{X} + \text{LayerNorm}_{\text{attn}}(a) \\
\text{output} &= \mathbf{h} + \text{LayerNorm}_{\text{ffn}}\left(\text{FFN}(\mathbf{h}, \theta_{\text{ffn}})\right),
\end{aligned}
\tag{1}
$$

where Attn() refers to the usual self-attention mechanism. The tokens of each modality in $\mathbf{X}$ are processed altogether at each attention layer.

## 3.2 MoT Attention

Contrary to the vanilla attention setting, where each token in the sequence is processed using the same weight matrix, in the MoT approach, as proposed in Liang et al. (2025), parameters are decoupled across modalities. However, the self-attention operation is still performed on the whole token sequence. Formally, for each $m \in \mathcal{M}$, the attention layer is equipped with dedicated projection matrices, $W_Q^m, W_K^m, W_V^m$, that are used to process each modality *independently* of the others to obtain the query, keys, and value matrices. Let $\mathbf{x}$ be decomposed as follows,

$$\mathbf{X} = \{\underbrace{\mathbf{x}_1, \ldots, \mathbf{x}_t}_{\text{text}}, \underbrace{\mathbf{x}_{t+1}, \ldots, \mathbf{x}_{t+k}}_{\text{audio}}, \underbrace{\mathbf{x}_{t+k+1}, \ldots, \mathbf{x}_n}_{\text{image}}\} \tag{2}$$

where each modality is located after the other. Note that the following also generalizes to interleaved situations. Then $\mathbf{x}_{1:t}$ is processed using $W_Q^{text}, W_K^{text}, W_V^{text}$, $\mathbf{x}_{t+1:t+k}$ is processed using $W_Q^{audio}, W_K^{audio}, W_V^{audio}$ and $\mathbf{x}_{t+k+1:n}$ is processed using $W_Q^{image}, W_K^{image}, W_V^{image}$. One then obtains,

$$Q = \texttt{Concat}([Q^{text}, Q^{audio}, Q^{image}])$$
$$K = \texttt{Concat}([K^{text}, K^{audio}, K^{image}]) \tag{3}$$
$$V = \texttt{Concat}([V^{text}, V^{audio}, V^{image}])$$

where the original sequence order is kept. Then, self-attention is performed on the whole sequence,

$$A = \texttt{softmax}\left(\frac{QK^\top}{\sqrt{d_k}}\right)V. \tag{4}$$

Let us denote, $I_m = \{i : m_i = m\}$ and $\mathbf{X}_m = \{x_i : i \in I_m\}$, then one obtains the per-modality output as

$$O_m = A_{I_m} W_O^m. \tag{5}$$

Following Liang et al. (2025) let us denote this entire processing as,

$$\texttt{GlobalAttn}(\mathbf{x}, \{\theta_{attn}^m\}_{m\in\mathcal{M}}) = \left(\texttt{softmax}\left(\frac{QK^\top}{\sqrt{d_k}}\right)V\right)W_O^{m_i} \tag{6}$$

Then, modality-specific LayerNorm and FFN are applied to each $O_m$ as described in equation equation 1. For a token $i$, this writes,

$$a = \texttt{GlobalAttn}\left(\mathbf{X}, \{\theta_{attn}^m\}_{m\in\mathcal{M}}\right)$$
$$\mathbf{h}_i = \mathbf{x}_i + \text{LayerNorm}_{attn}^{m_i}(a_i) \tag{7}$$
$$\text{output}_i = \mathbf{h}_i + \text{LayerNorm}_{ffn}^{m_i}\left(\text{FFN}_{m_i}(\mathbf{h}_i, \theta_{ffn}^{m_i})\right)$$

## 3.3 Mixture-of-LoRA (MoL)

In Fig. 1 we display an overview of the mechanisms involved in a MoL layer. Let us denote the weights of a pretrained LLM as $\bar{\theta}$. Our proposed method leverages the LLM's pretrained weight matrices to fine-tune them for multimodal inputs efficiently. Similarly to section 3.2, let $\mathbf{x}$ be decomposed as shown in equation 2.

As in the standard setting, one will obtain the usual query, key, and value representations, $Q, K, V$ using $\bar{W}_Q, \bar{W}_K, \bar{W}_V \in \bar{\theta}$. We propose using modality specific LoRA adapters, $\{(A_Q^m, B_Q^m)\}_{m\in\mathcal{M}}$, $\{(A_K^m, B_K^m)\}_{m\in\mathcal{M}}$ and $\{(A_V^m, B_V^m)\}_{m\in\mathcal{M}}$ while keeping the original layers frozen. Let $r$ designate the chosen rank of the LoRA adapters and $d$ the hidden dimension of the model, then $A_L^m \in \mathbb{R}^{r\times d}, B_L^m \in \mathbb{R}^{d\times r}$ for $L \in \{K, V, Q\}$. For each modality $m \in \mathcal{M}$, we compute,

$$\tilde{Q}^m = \mathbf{X}_m B_Q^m A_Q^m,$$
$$\tilde{K}^m = \mathbf{X}_m B_K^m A_K^m, \tag{8}$$
$$\tilde{V}^m = \mathbf{X}_m B_V^m A_V^m.$$

---

**Algorithm 1** Mixture-of-LoRA (MoL) Layer

---

1: Let $\mathbf{x} = (\mathbf{x}_1, \ldots, \mathbf{x}_n)$ be the input sequence, where $\mathbf{x}_i \in \mathbb{R}^d$ and $m_i \in \{text, image, speech\}$ is the modality of token $\mathbf{x}_i$.
2: Let $\mathcal{M} = \{text, image, speech\}$ be the set of modalities.
3: Let $W_Q, W_K, W_V \in \theta$ denote the frozen layer of the fine-tuned LLM.
4: Let $(A_Q^m B_Q^m), (A_K^m B_K^m), (A_V^m B_V^m), (A_O^m B_O^m)$ denote the LoRA adapters for modality $m$, $r$ the corresponding rank and $\alpha$ the scaling factor.
5: Let $\text{FFN}_m$ denote the FFN networks equipped with modality $m$ MoL adapter.
6: **for** each modality $m \in \mathcal{M}$ **do**
7:      $I_m \leftarrow \{i : m_i = m\}$          ▷ Indices of tokens for modality $m$
8:      $X_m \leftarrow \{x_i : i \in I_m\}$          ▷ Group tokens by modality
9:      $\tilde{Q}_m \leftarrow \mathbf{X}_m B_Q^m A_Q^m$          ▷ Modality-specific LoRA adapters
10:     $\tilde{K}_m \leftarrow \mathbf{X}_m B_K^m A_K^m$
11:     $\tilde{V}_m \leftarrow \mathbf{X}_m B_V^m A_V^m$
12: **end for**
13: $\tilde{Q} \leftarrow \bigcup_{m \in \mathcal{M}} \tilde{Q}_m, \tilde{K} \leftarrow \bigcup_{m \in \mathcal{M}} \tilde{K}_m, \tilde{V} \leftarrow \bigcup_{m \in \mathcal{M}} \tilde{V}_m$      ▷ Aggregate LoRA representations
14: $\mathbf{Q}_{\text{MoL}} \leftarrow XW_Q + \frac{\alpha}{r}\tilde{Q}, \mathbf{K}_{\text{MoL}} \leftarrow XW_K + \frac{\alpha}{r}\tilde{K}, \mathbf{V}_{\text{MoL}} \leftarrow XW_V + \frac{\alpha}{r}\tilde{V}$
15: $A \leftarrow \text{softmax}\left(\frac{QK^T}{\sqrt{d_k}}\right) V$          ▷ Global self-attention
16: $O \leftarrow AW_O$
17: **for** each modality $m \in \mathcal{M}$ **do**
18:     $\tilde{O}_m \leftarrow A_{I_m} B_O^m A_O^m$          ▷ Modality-specific LoRA projection
19:     $O_m \leftarrow O_m + \frac{\alpha}{r}\tilde{O}_m$          ▷ Modality-specific output projection
20:     $H_m \leftarrow X_m + \text{LayerNorm}_{\text{attn}}^m(O_m)$          ▷ Residual connection and layer norm
21:     $F_m \leftarrow \text{FFN}_m(H_m)$          ▷ Feed-forward network equipped with MoL adapters
22:     $Y_m \leftarrow H_m + \text{LayerNorm}_{\text{ffn}}^m(F_m)$          ▷ Residual connection and layer norm
23: **end for**
24: **return** $\{Y_m : m \in \mathcal{M}\}$          ▷ Return transformer layer outputs

---

Then, one concatenates the obtained representations,

$$\tilde{Q} = \text{Concat}([\tilde{Q}^{text}, \tilde{Q}^{audio}, \tilde{Q}^{image}]),$$
$$\tilde{K} = \text{Concat}([\tilde{K}^{text}, \tilde{K}^{audio}, \tilde{K}^{image}]), \quad (9)$$
$$\tilde{V} = \text{Concat}([\tilde{V}^{text}, \tilde{V}^{audio}, \tilde{V}^{image}]).$$

Those representations are then added to the representations obtained from the frozen weights of the LLM, to obtain $\mathbf{Q}_{\text{MoL}}, \mathbf{K}_{\text{MoL}}, \mathbf{V}_{\text{MoL}}$. Note that a similar process is performed on the output matrix, $W_O$. One can also include per-modality LayerNorm and LoRA adapters to the FFN networks. While including per-modality LayerNorm does not induce any significant computational overhead, replacing the pretrained FFN network with a per-modality module, as done in (Liang et al., 2025), would require retraining a significant share of the parameters of the LLM. Thus, we consider MoL adapters to the attention matrices $W_Q, W_K, W_V, W_O$ and the FFN network, and include modality-specific LayerNorm modules for each attention layer. See algorithm 1 for a description of the overall process.

### 3.4 INPUT REPRESENTATION AND TOKENIZATION

We unify all modalities into a common sequence representation to enable the LLM to process visual and audio information. We first encode images and audio into sequences of discrete tokens using pretrained encoders. Specifically, each vector from the discrete codebook of the image and audio encoders is treated as a new, special token that is added to the LLM's tokenizer vocabulary. The embeddings for these new modality-specific tokens are initialized from the corresponding vector representations in their original modality's codebook. A linear projection layer is used to map the dimension of the codebook vectors to the hidden dimension of the LLM. Additionally, we include tokens delimiting a modality's tokens in the LLM's vocabulary. The intent is to inform the model when it needs to predict a specific modality, e.g. `,<\img>` or `<speech>,<\speech>`.

For instance, let $\mathbf{x}$ be an image and $f(\cdot)$ be the image encoder that maps $\mathbf{x}$ to a sequence of indices from its codebook $Q$. If the resulting sequence of indices is $f(x) = \{i_1, i_2, \ldots, i_N\}$, the final input sequence fed to the LLM is constructed as

$$\texttt{<image\_token\_}i_1\texttt{><image\_token\_}i_2\texttt{>}\ldots\texttt{<image\_token\_}i_N\texttt{>}.$$

Each $\texttt{<image\_token\_}i_j\texttt{>}$ corresponds to a unique vector in the LLM's expanded embedding matrix. This method allows the LLM to process the other modalities as if they were a sequence of text while retaining the rich, pretrained representations from the original encoder. Moreover, this approach not only allows the pretrained LLM to *understand* other modalities than text, but it also enables the model to *generate* multimodal outputs.

## 4 EXPERIMENTS

In the present section, we discuss the experiment settings, including hyperparameter settings and training dataset, and then we discuss the results.

### 4.1 EXPERIMENTAL SETTINGS

**Datasets** For the Chameleon setting, we rely on three datasets to train our model, MS-COCO (Lin et al., 2015), Laion-400M (Schuhmann et al., 2021), and Flickr-30k (Young et al., 2014). We evaluated our model's performance using validation losses on held-out sets MS-COCO and Flickr30k. In particular, following previous work (Liang et al., 2025), we use the Karpathy test split of MS-COCO and Flickr30k as the validation sets. For the setting including all three modalities, we also include the English split of the MultiLingual Librispeech dataset (Pratap et al., 2020) in training, and rely on a held-out split for validation. Compared to model training a multimodal foundational model from scratch, e.g., Chameleon (Team, 2024) or MoT (Liang et al., 2025), in our setup, the LLM does not need to see many text tokens, as the pretrained LLM has already been trained on a significant share of text tokens. Our fine-tuning dataset contains a mixture of text and image tokens, representing respectively 5% and 95% of total tokens for the Chameleon setup. For the three-modality setting, we sample across datasets so that image tokens represent 65% of total tokens, speech tokens 30%, and text tokens 5%. We select more image tokens than speech tokens because the image encoder involves a larger vocabulary than the speech extractor, as discussed in the following paragraph.

**Model hyperparameters** For the image and audio feature extractors, we respectively rely on the VQ-VAE made available by Team (2024) and DinoSR (Liu et al., 2023a), also available online. The former has a *vocabulary* size of 8192 image tokens, and the latter has 256 audio tokens that are added to the LLM's vocabulary (see section 3.4). Our experiments are conducted using Qwen 2-0.5B (Yang et al., 2024b), Llama-3.2-1B, and Llama-3.2-3B (Grattafiori et al., 2024) as the base LLMs. We fine-tune them by replacing their attention layers with our proposed MoL Layer. We also consider per-modality LayerNorm, but omit MoL augmented FFNs and discuss their addition in section 5.2. LoRA adapters' rank and $\alpha$ are scaled with the size of the base model as described in section A.2. We optimize the model's weights using AdamW (Loshchilov & Hutter, 2019). In both Chameleon and three-modality settings, we set different learning rates for each modality's LoRA adapters as we observed that it helped stabilize training and avoid gradient explosion. We provide extended detail on the optimizer's hyperparameters in section A.1 in the appendix. For the 1B setup, we rely on a learning rate scheduler with $10,000$ warm-up steps progressively increasing to the target learning rate, and a cosine decay for the remaining training steps. We fix the context size to 2048 for all three setups, and ensure that samples are not truncated in the middle of an audio or image token sequence. Token sequences are randomly ordered with respect to the modalities, i.e., for a sample containing an image and a textual description of the image, we randomly select whether image or text tokens come first in the sequence.

We clip gradients to $0.5$ and use an effective batch size of 256 for the 1B and 3B settings, thus providing the model with an average of 500k multimodal tokens per optimization step. We select an effective batch size of 128 for Qwen-2 0.5B fine-tuning. We train Llama-3.2 1B and 3B using 32 H100 Nvidia GPUs and 8 H100 Nvidia GPUs for Qwen-2 0.5B. Our main experiment focused on fine-tuning Llama-3.2 1B for 35,000 steps. The Llama-3.2 3B and Qwen-2 0.5B models were trained with a comparatively smaller number of steps, serving as proof-of-concept demonstrations to

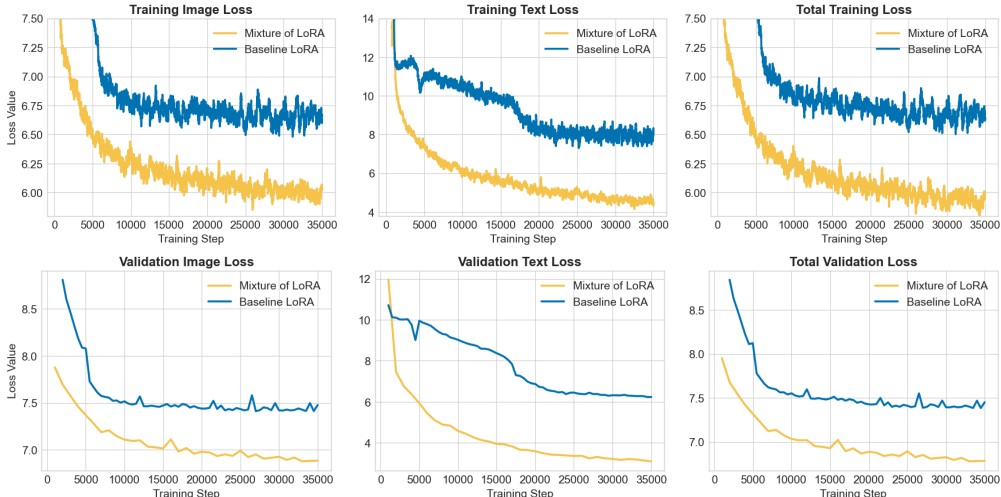

Figure 2: **Loss curves for the Chameleon setting (Llama-3.2 1B).** We observe that the training losses consistently decrease as training progresses. This demonstrates the ability of our MoL adapters to learn to process multimodal tokens effectively. Similarly, validation losses follow a similar pattern to the training loss. For frugality reasons, we are unable to train the model further, but these loss curves indicate that the model is still underfit and could improve further.

validate our approach across different model sizes and architectures while prioritizing computational efficiency. See appendix A for extensive details on all three setups.

## 4.2 RESULTS

To demonstrate the relevance of MoL as a multimodal fine-tuning method, we fine-tune several models of different sizes, from 0.5B parameters to 3B, and compare their performance to a baseline LoRA model where all modalities share a LoRA adapter. We trained the baseline LoRA models with identical relevant hyperparameters to the MoL model. We provide in figure 6, 2 and 7 the loss curves for Qwen-2 0.5B (Yang et al., 2024b), Llama-3.2 1B and Llama-3.2 3B (Grattafiori et al., 2024) respectively.

We observe a consistent pattern across our setups: models fine-tuned with MoL adapters show a steady decrease in loss. The loss curves for Llama-3.2 1B and 3B, augmented with MoL adapters, exhibit a consistent downward trend, suggesting that a further decrease in loss would likely occur with additional training steps. Conversely, the Qwen-2 0.5B model's image loss rapidly saturates after 3000 steps, even as its text loss improves. This phenomenon suggests that increasing the MoL adapters' rank, particularly those for the image modality, might further enable performance gains. In comparison, the baseline approach of fine-tuning with a single LoRA adapter reveals a modality conflict. Initially, the models appear to learn, as evidenced by a downward loss trend. However, this progress plateaus, and modality losses start displaying different trends. The image token loss rapidly hits a performance ceiling and stagnates across all setups. While the text loss does not saturate, its optimization path is unstable and fails to decrease consistently. This behavior suggests that a single, shared adapter is forced to learn competing and non-transferable representations for vision and language, creating a representational bottleneck. The adapter's updates are pulled in conflicting directions, leading to saturation on one modality and erratic performance on the other.

## 5 DISCUSSION

To further investigate the relevance of our proposed approach, we investigate three additional settings. First, we consider a three-modality setting where the pretrained LLM not only learns to receive image and text but also speech modality tokens. Second, we compare the performance of a model with and

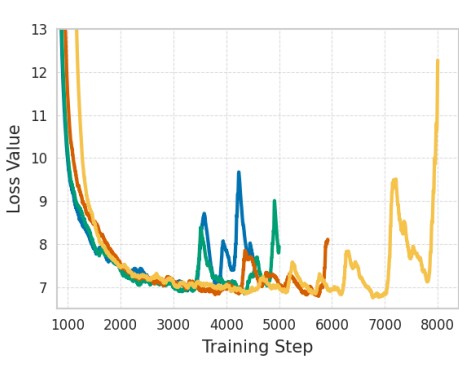 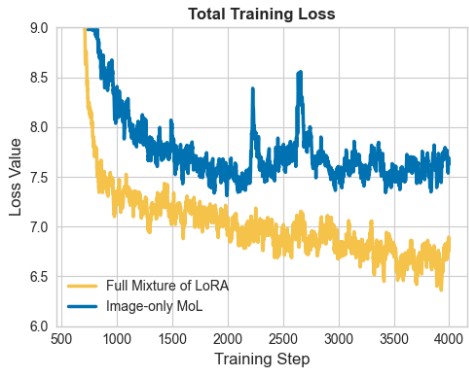

(a) **MoL adapters on FFN networks**.  (b) **Ignoring MoL adapters for text tokens**.

Figure 3: **Ablation for the Llama-3.2 1B Chameleon setup.** (a): Training loss curves when including MoL adapters on the FFN networks with various learning rate and optimizer parameters. We observe that in all tested setups, the total loss (primarily driven by the image loss) diverges after a few thousand steps. (b): Training loss curves comparing the full MoL setup (adapters for both image and text tokens) versus the image-only MoL setup. The full MoL setup displays a smooth, downward trend. At the same time, the image-only MoL model quickly saturates and exhibits an erratic trend, suggesting that text token adaptation is critical for successful convergence.

without MoL adapters for the FFN networks. Finally, we investigate whether including a text MoL adapter is necessary for efficiently learn to include additional modalities into an existing model.

## 5.1 THREE MODALITY SETTING

To further evaluate the abilities of our approach to fuse new modalities into pretrained LLMs, we fine-tune Llama-3.2 1B with three distinct modalities: text, speech, and images. As shown by the converging loss curves in Fig. 4 (Appendix B.1), our model successfully learns to integrate both additional modalities. Notably, due to the significant variance in audio sample length within the training dataset, we employed curriculum learning for the speech modality, starting with shorter audio inputs and progressively increasing their size.

The speech modality presents the most significant challenge for integration into the pretrained model. Specifically, the audio-specific loss often exhibits volatile peaks during the initial training stages, severely hindering stability. We hypothesize that this instability stems from the audio feature extractor being less expressive than its image counterpart. This reduced expressiveness is evident in its significantly smaller vocabulary (i.e., token set), likely leading to increased ambiguity or confusion among audio tokens.

## 5.2 IMPACT OF THE MoL ADAPTERS ON THE FEED-FORWARD NETWORKS

While our baseline configuration applies MoL adapters solely to the attention mechanism's query ($W_Q$), key ($W_K$), value ($W_V$), and output ($W_O$) projections, we also evaluate an extended setup. In this configuration, we integrate MoL into the feed-forward network (FFN) layers to investigate the benefits of increasing the model's learning capacity. Overall, we iterated over a high number of hyperparameter settings and observed that in all settings, including FFN, allowed the loss curves to rapidly decrease in the first steps compared to the *attention-only* MoL setting. However, it caused significant loss divergence after some time, even with careful learning rate scheduling and an optimizer's hyperparameter tuning. We display in Fig. 3a the loss curves of different model experiments we conducted using Llama-3.2 1B as the base model on the Chameleon setup. In particular, this loss divergence mainly occurs on image tokens, while text tokens are preserved from this loss explosion.

### 5.3 MoL ADAPTER FOR NON-TEXT MODALITIES

Given the base model's extensive text pre-training, we investigated the necessity of a dedicated text MoL adapter during multimodal fine-tuning. Fig. 3b displays the loss curves for Llama-3.2 1B in the Chameleon setting, comparing a model equipped with only image MoL adapters (Image-only MoL) against one using both text and image adapters (Full Mixture-of-LoRA) under identical hyperparameters. While initial loss reduction is observed in both setups for the first ~2000 steps, the full MoL configuration converges faster and more stably. In contrast, the image-only model's loss plateaus and becomes erratic after this point, indicating that text token adaptation is critical for maintaining stable and effective multimodal training.

An analysis of per-modality loss (Fig. 5, Appendix B.2) reveals that the model struggles significantly to predict text tokens when augmented solely with image MoL adapters. We hypothesize that this performance gap stems from a representational asymmetry in the image-only configuration. In this setup, image tokens are adaptively transformed by the MoL layers, while text tokens are processed solely by the frozen, pre-trained weights of the LLM. This forces the image adapters to bear the burden of cross-modal alignment, requiring them to map visual features into a fixed and potentially suboptimal textual representation space. Including text adapters resolves this by introducing complementary transformation: the text adapters learn to condition the token representations for optimal fusion, thereby allowing the image adapters to focus on their primary task of modality-specific feature representation.

## 6 CONCLUSION

The present work proposes Mixture-of-LoRA (MoL), a novel, parameter-efficient method for equipping pretrained large language models with multimodal capabilities. Our experiments in text-image and text-image-audio settings demonstrate that MoL enables LLMs to understand and generate multimodal data effectively. MoL significantly outperforms a standard LoRA baseline with negligible computational overhead. By leveraging the extensive knowledge of the base LLM, our approach circumvents the need for training a multimodal model from scratch, thus presenting a computationally efficient alternative to existing methods (Team, 2024; Liang et al., 2025).

**Limitations and future work**  Our current framework requires modality-specific feature extractors with a discrete latent space. A promising direction for future work is to investigate the applicability of MoL to understanding-only tasks that use continuous representations from encoders like WavLM (Chen et al., 2021) for audio or CLIP (Radford et al., 2021) for vision.
Furthermore, our evaluation focused on an agnostic next-token prediction objective. We leave the assessment of MoL's effectiveness on specific downstream multimodal tasks–such as Visual Question Answering (VQA), Visual Reasoning, or Speech-to-Speech Translation–as an important avenue for future research.

**Reproducibility statement**  This research is partially reproducible. We have provided all the necessary information, including hyperparameters, experimental settings, and datasets, in section 4 to allow an independent researcher to replicate our findings. Moreover, we provide in algorithm 1 all the necessary information to implement our approach and the base code in Python in the supplementary material. The provided code works with most large language models loaded from the `transformers` library and would only require minor adjustments to work on different architectures. However, we have not released the training code to generate the results. This means that a researcher would have to re-implement the training pipeline, which could introduce variations and make an exact replication challenging. Therefore, while the core methodology can be followed, a direct, bit-for-bit reproduction of the results is not possible without access to the original training code.

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

# A  EXPERIMENTAL SETTINGS

## A.1  OPTIMIZER

As described in section 4.1, we opt for AdamW (Loshchilov & Hutter, 2019) as the optimizer to update the weights during training. We select different learning rates for each model size for each modality parameter as we observe that it enables more stable loss convergence. We provide in table 1 details on each hyperparameter for the different training setups.

Table 1: **Optimizer hyperparameters**.

| Base model | training setting | base lr | img lr | text lr | speech lr | $(\beta_1, \beta_2)$ |
|---|---|---|---|---|---|---|
| Qwen-2 0.5B | Chameleon | $6.e^{-5}$ | $5.e^{-5}$ | $5.e^{-5}$ | N/A | $(0.97, 0.999)$ |
| Llama-3.2 1B | Chameleon | $4.e^{-4}$ | $2.e^{-4}$ | $5.e^{-5}$ | N/A | $(0.97, 0.999)$ |
| Llama-3.2 1B | Three-modality | $2.e^{-4}$ | $1.e^{-4}$ | $9.e^{-5}$ | $6.e^{-5}$ | $(0.975, 0.999)$ |
| Llama-3.2 3B | Chameleon | $1.e^{-4}$ | $1.e^{-4}$ | $5.e^{-5}$ | N/A | $(0.97, 0.999)$ |

## A.2  MIXTURE OF LoRA

Depending on the base model's size and the training setting, we select different values for the LoRA rank $r$ and the scaling factor $\alpha$. We display the chosen values in table 2 for each setup.

Table 2: **LoRA hyperparameters**.

| Base model | training setting | $r$ | $\alpha$ |
|---|---|---|---|
| Qwen-2 0.5B | Chameleon | 16 | 16 |
| Llama-3.2 1B | Chameleon | 64 | 64 |
| Llama-3.2 1B | Three-modality | 64 | 64 |
| Llama-3.2 3B | Chameleon | 64 | 64 |

## A.3  TRAINABLE PARAMETERS

We provide details on trainable parameters share in table 3.

Table 3: **Trainable parameters**.

| Base model | training setting | trainable parameters | parameter count | trainable share |
|---|---|---|---|---|
| Qwen-2 0.5B | Chameleon | 177M | 535M | 33.1% |
| Llama-3.2 1B | Chameleon | 306M | 1.27B | 23.9% |
| Llama-3.2 1B | Three-modality | 321M | 1.29B | 24.8% |
| Llama-3.2 3B | Chameleon | 492M | 3.3B | 14.8% |

# B EXPERIMENTS

## B.1 THREE MODALITIES

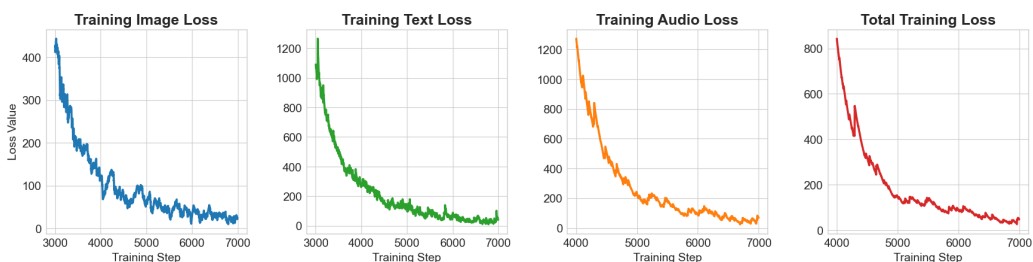

Figure 4: **Loss curves for the Three-modality setting (Llama-3.2 1B).** We observe that all three modality-specific loss curves display a similar decreasing shape. The model augmented by the MoL adapters can learn to fuse simultaneously three modalities as its total loss displays a downward curve.

For visibility, we do not include in the graphs in figure 4 the first training steps; nevertheless, we observe that early training is more erratic when all three modalities are included. In particular, even when controlling the gradient norm with gradient clipping to small values, e.g. $0.5$, loss can explode for both image and audio tokens. This behavior is only observed in the early stage of training and progressively disappears after 3000 training steps.

## B.2 ADDITIONAL DETAILS ON MoL ADAPTER FOR NON-TEXT MODALITIES

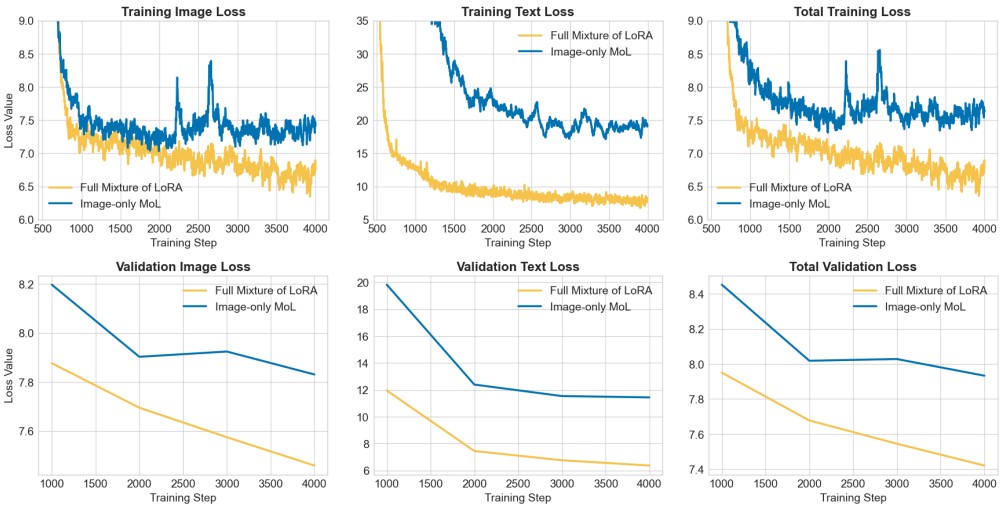

Figure 5: **Loss curves between Llama-3.2 1B with MoL adapters for image and text vs image only.** In contrast to the full MoL model, the image-only model's loss quickly plateaus and becomes erratic after ~2000 steps. Also, we observe that the image-only MoL model struggles to predict text tokens.

### B.3 OTHER PARAMETER COUNT SETTINGS

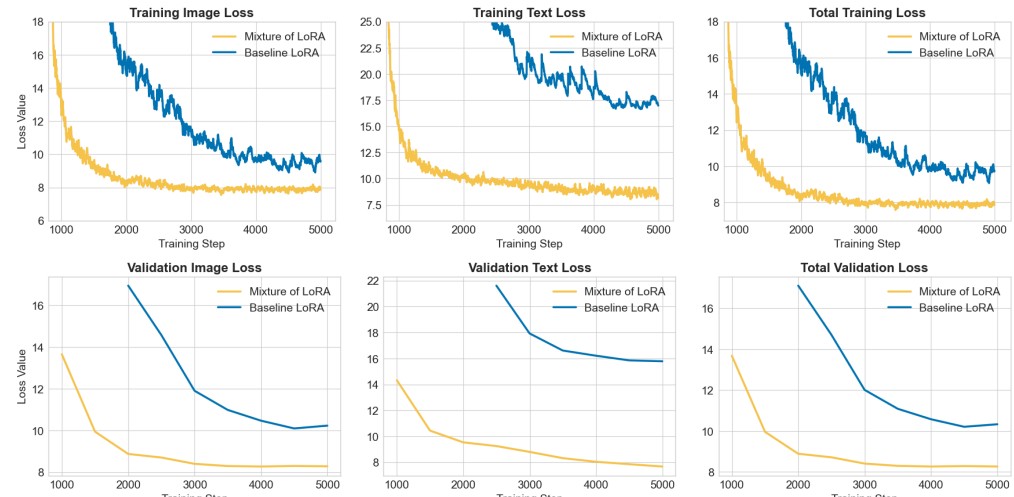

Figure 6: **Training losses for Qwen 2-0.5B.** We observe a similar pattern as bigger models, where all modality losses consistently decrease for both MoL and the baseline LoRA model. However, the decrease requires more tokens to reach a satisfactory value. Contrary to the bigger models' setting, we use a similar learning rate for all modalities' parameters, but rely on the same number of tokens per optimization steps as for the 1B setup. Regarding the baseline LoRA, the loss curves are more erratic, as shown in figure 2.

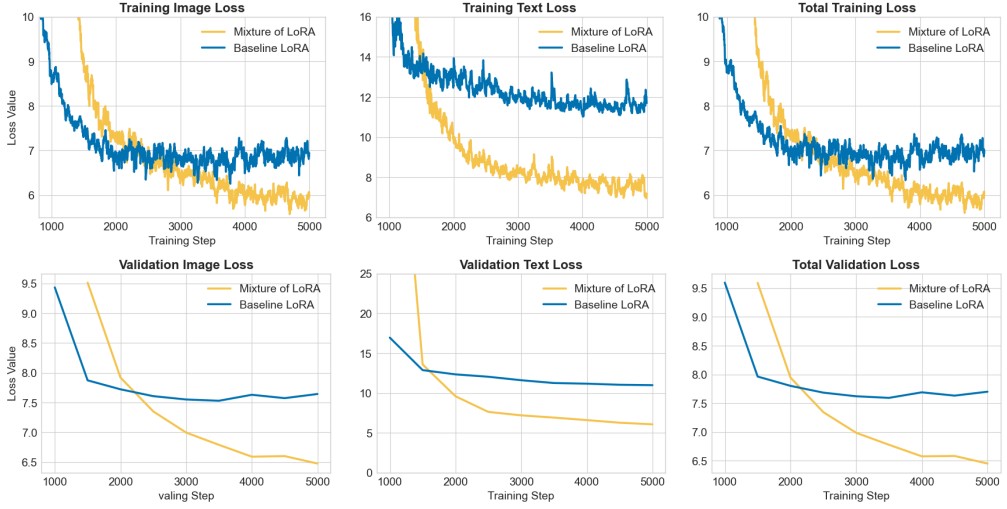

Figure 7: **Training losses for Llama-3.2 3B.** We observe that the MoL model and the baseline LoRA display decreasing learning curves during the first training steps. However, as training progresses after step ~3000, the LoRA model appears to plateau and even shows an upward trend while the MoL model continues to decrease. Notably, the loss for the LoRA model decreases faster during the first steps, but its limited learning capacity in comparison to the MoL model prevents it from further improving after step ~2000.

## C    LLMS USAGE

During the preparation of this manuscript, Large Language Models (LLMs) were consulted for the limited purpose of refining language and style. All intellectual contributions, analyses, and conclusions are entirely the work of the authors.

