# OpenReview forum: "Scalable Multimodal Fine-tuning for Foundation Models via Mixture-of-LoRA"
_ICLR.cc/2026/Conference — Submitted to ICLR 2026_

### Official Review · Reviewer_gHzZ · 2025-10-24

**Soundness:** 2
**Presentation:** 2
**Contribution:** 2
**Rating:** 2
**Confidence:** 3

**Summary:**

This paper proposes **Mixture-of-LoRA (MoL)**, a parameter-efficient approach to extend *text-only* large language models (LLMs) to **multimodal understanding and generation** (text–image–audio).
The key idea is to insert modality-specific LoRA adapters into the attention projections ((W_Q, W_K, W_V, W_O)) of a frozen pretrained LLM.
Each modality (e.g., text, image, audio) gets its own adapter that produces (Q, K, V) representations. These are concatenated in sequence order and added, scaled by (a / r), to the frozen attention projections before shared global self-attention.
This design borrows the intuition of Mixture-of-Transformers (MoT), separate per-modality projections with shared global attention, but replaces full per-modality parameters with low-rank adapters, achieving better scalability and efficiency.
Experiments adapt Qwen-2-0.5B, Llama-3.2-1B, and Llama-3.2-3B using image and audio tokenizers (VQ-VAE for images and DinoSR for audio). The models are trained on mixed-modality next-token prediction tasks using LAION-400M, MS-COCO, Flickr30k, and MLS-en datasets.
Results show that MoL achieves steadier and lower loss curves than single shared LoRA baselines, indicating improved stability and cross-modality learning.

**Strengths:**

1. The paper provides  a precise, simple recipe to make a frozen text LLM multimodal using per-modality LoRA on attention projections with global self-attention. The MoL diagram and Algorithm 1 are easy to follow.

2. Across Qwen-0.5B, Llama-1B, and Llama-3B, MoL reduces losses more steadily than a single shared LoRA, suggesting reduced cross-modality interference.

3. It finds text adapters matter: using only image adapters harms stability and text prediction; symmetric adaptation helps cross-modal alignment

**Weaknesses:**

1. This paper lacks of downstream task evaluation. Claims of “understand and generate” are not backed by standard tasks (e.g., COCO captioning, VQAv2/GQA, NLVR2, Text-to-Image FID/IS/CLIP-score, ASR/speech-to-text/XTTS). The show of  loss trends is not enough.

2. This paper also missing some related  baselines. The comparison is mainly to a single shared LoRA. Authors could consider more baselines like: (1) Strong connector baselines (e.g., CLIP/ViT encoders + projection to LLM with PEFT). (2) MoT-style finetuning of a frozen LLM via lightweight per-modality projections (if feasible). (3) Other PEFT variants (IA³, LoRA-plus, DoRA) or multi-adapter routing. It is hard to figure out the benefit of multi-lora for multimodal learning without task specific baselines.

3. The authors report FFN-adapter divergence but this hasn't been  probed (e.g., optimizer dynamics, scale mismatch, rank/α sweeps, gradient stats, adapter placement sensitivity).

**Questions:**

1. Can you report task metrics to substantiate “understand & generate”? Suggested minimal suite:
Text↔Image: COCO captioning (CIDEr/SPICE), NoCaps, VQAv2/GQA; text-to-image (e.g., coarse CLIP-score).
Audio: MLS-en ASR WER; spoken captioning or speech-grounded QA.


2. Which attention projections contribute most (Q vs K vs V vs O)? Any benefit from placing adapters only in deeper layers or at cross-modal “fusion” layers identified by probing?

3. Can you provide FLOPs, tokens/sec, and latency overhead vs single-LoRA and connector baselines for the 1B/3B models? The trainable-share table is helpful but efficiency numbers would be more actionable.

---

> ### Author Response · Authors · 2025-11-21
> **Response to Reviewer gHzZ**
>
> We would like to thank **Reviewer gHzZ** for their valuable comments, which will enable us to improve the quality of our work. Allow us to comment and respond to a few points raised by them.
>
>
> **Reviewer gHzZ**: *This paper lacks of downstream task evaluation (...) The show of loss trends is not enough.*
>
> The core contribution of our work is a parameter-efficient multimodal pretraining strategy that enables generation. We position our work alongside pretraining-focused methods, such as [1], which also primarily assess performance via per-modality loss. They compare two concurrent methods, e.g., a dense transformer (Chameleon) and a MoE-based architecture, which they train themselves. We compare MoL to a single, shared LoRA adapter. This is the most equivalent parameter-efficient baseline for multimodal pretraining.
>
> We acknowledge the importance of evaluating downstream tasks. We are actively working to produce relevant benchmarks against existing fine-tuning methods. However, we note that most comparable methods [2,3,4] are typically fine-tuned directly on those downstream tasks (e.g., VQA, Captioning) and often rely on Instruct-like models, making a direct, apples-to-apples comparison difficult without extensive retraining. Our primary focus remains on multimodal generation enabled by pretraining.
>
> **Reviewer gHzZ**: *This paper also missing some related baselines. (...)Other PEFT variants (IA³, LoRA-plus, DoRA) or multi-adapter routing.*
>
> While some alternative baselines are indeed relevant (and we will try to include as many as possible), we emphasize that not all suggestions made by **Reviewer gHzZ** can be included in our setup. For instance, the approach proposed in the present work necessitates a discrete representation of multimodal inputs (this is why we resort to VQ-VAE, DinoSR, or Mimi to represent image/audio). Thus, the suggestion to include "*CLIP/ViT encoders + projection to LLM with PEFT*" would not allow for generating multimodal tokens, as MoL does (as discussed in Section 3.4).
> Regarding the alternative PEFT methods, we will include some of them in ablation before the end of the discussion period (in particular LoRA-plus and DoRA).
>
> **Reviewer gHzZ**: *Which attention projections (...) layers identified by probing?*
>
> Preliminary analysis of adapter norms suggests that the network utilizes MoL adapters uniformly across depth, implying that multimodal alignment requires capacity throughout the network, not just at "fusion" layers.
> Given that the overall new task to which the base model must adapt is quite complex (new multimodal vocabulary understanding and generation), we believe that adding LoRA adapters only to some layers will likely weaken the fine-tuning method's ability. Nevertheless, we will conduct an ablation study to demonstrate this and include the results in the manuscript. We thank **Reviewer gHzZ** for this comment, as it will be of great interest to add this analysis to our work.
>
> Regarding component-wise sensitivity, interestingly, we observe distinct behaviors in the Query projections. The norms of MoL adapters applied to $W_Q$​ are consistently higher (approx. +12-30% relative to K/V/O) across modalities. This could suggest that the model relies heavily on modifying the query to route attention to the correct modality context (image vs. text tokens). We are formalizing this analysis for the final manuscript.
>
> **Reviewer gHzZ**: *Can you provide FLOPs, (...) actionable.*
>
> - Llama 1B-MoL on H100 (mixed precision): 3.39 samples/sec (average over 40k steps)
> - Llama 1B-Baseline LoRA on H100 (mixed precision): 5.63 samples/sec (average over 40k steps)
> - Llama 3B-MoL on H100 (mixed precision): 3.18 samples/sec (average over 5k steps)
> - Llama 3B-Baseline LoRA on H100 (mixed precision): 4.02 samples/sec (average over 5k steps)
>
> - Llama 1B Baseline LoRA: 340.48 TFLOPs/s
> - Llama 1B MoL, 204.92 TFLOPs/s
> - Llama 3B Baseline LoRA, 634.46 TFLOPs/s
> - Llama 3B MoL, 502.50 TFLOPs/s
>
> We will provide some discussion on this topic in the manuscript before the end of the discussion phase.
>
> [1] Mixture-of-transformers: A sparse and scalable architecture for multimodal foundation models. Weixin Liang, et al. in TMLR, 2025.
>
> [2] Visual instruction tuning. Haotian Liu, Chunyuan Li, Qingyang Wu, and Yong Jae Lee. NeurIPS 2023.
>
> [3] Qwen-VL: A Versatile Vision-Language Model for Understanding, Localization, Text Reading, and Beyond. Jinze Bai, et al. 2023
>
> [4] Sharegpt4v: Improving large multimodal models with better captions. Lin Chen, et al. 2023.

---

### Official Review · Reviewer_rJ7J · 2025-10-31

**Soundness:** 2
**Presentation:** 2
**Contribution:** 2
**Rating:** 2
**Confidence:** 3

**Summary:**

This paper introduces Mixture-of-LoRA (MoL), a scalable and parameter-efficient approach for multimodal fine-tuning of large language models (LLMs).The key idea is to insert modality-specific low-rank adapters (LoRA modules) into the attention and output projection layers while maintaining shared global attention for cross-modal fusion.

Experiments are conducted on Qwen-2 (0.5B) and Llama-3.2 (1B, 3B) models, covering text, image, and audio modalities.
The method demonstrates better convergence and multimodal alignment compared to shared LoRA baselines, particularly for small and medium-sized models.

However, the use of mixed base model families (Qwen and Llama) complicates interpretation — improvements may partly arise from differences in tokenizer design, pretraining corpus, or training stability rather than from the MoL architecture itself.
Furthermore, MoL’s benefit diminishes as model size increases, and full-duplex (streaming) speech interaction remains unsupported.

**Strengths:**

1. MoL elegantly combines Mixture-of-Transformers and LoRA for efficient multimodal adaptation.

2. Achieves significant reduction in trainable parameters with comparable or better multimodal alignment.

3. Detailed ablation analysis — Addresses FFN instability, modality-specific LoRA effects, and learning-rate sensitivity.

**Weaknesses:**

1. Diminishing returns at larger scales. Gains drop from ~10% at 0.5B to <2% at 3B; **likely negligible at ≥8B or 33B**.

2. Mixed base models confound interpretation. Using both Qwen and Llama introduces architecture, tokenizer, and corpus differences; observed gains may be partially due to base-model factors rather than MoL itself. Without a Qwen-only or Llama-only scale study, scaling trends cannot be rigorously isolated.

3. Limited downstream evaluation. No quantitative benchmarks (VQA, captioning, ASR, etc.), relying solely on validation loss.

**Questions:**

1. Run two controlled single-family studies (Qwen-only and Llama-only) to isolate MoL’s true scaling behavior.

2. Include downstream quantitative tasks.

3. Investigate cross-family transfer (e.g., adapters trained on Qwen applied to Llama) to test modular generalization.

---

> ### Author Response · Authors · 2025-11-20
> **Response to Reviewer rJ7J**
>
> We are grateful to **Reviewer rJ7J** for their comments. Allow us to provide some additional information and answer a few interrogations on some of the points raised by them.
>
> **Reviewer rJ7J**: *Diminishing returns at larger scales. Gains drop from ~10% at 0.5B to <2% at 3B (...)*
>
> We respectfully clarify our position regarding the scaling behavior of MoL. While Figure 6 (Qwen 0.5B) shows a larger absolute loss reduction than Figure 7 (Llama 3B), we believe this observation is premature due to differences in experimental setup:
> - **Training Convergence**: The 3B model, having significantly more parameters, is known to require more steps to reach a comparable state of convergence than the 0.5B model, despite the token count per step. Direct comparison of two runs truncated at the same 5k steps is thus inappropriate.
> - **Loss Trend**: Crucially, the curves in Figure 7 clearly show the gap between MoL and the LoRA baseline is still actively widening at the 5k step mark. We hypothesize that given sufficient, proportional training time (e.g., 10~15k steps), the final convergence gap would be much more significant, demonstrating persistent benefits rather than diminishing returns.
>
> **Reviewer rJ7J**: *Mixed base models confound (...) scaling trends cannot be rigorously isolated.*
>
> We must clarify that a core objective of our experimental design was to demonstrate architectural versatility and robustness by successfully applying MoL across different foundational LLM families (Qwen-2 and Llama-3.2). This was a deliberate choice to show MoL is not an architecture-specific trick.
> The combination (Qwen 0.5B, Llama 1B/3B) was chosen to maximize the tested parameter range given our computational budget, as Qwen provided a strong, publicly available small-scale baseline not present in the Llama family.
>
> **Reviewer rJ7J**: *Investigate cross-family transfer (e.g., adapters trained on Qwen applied to Llama) to test modular generalization.*
>
> To our knowledge LoRA adapters **are highly dependent** on the architecture and the **specific weights** of the base model. Since different families of LLMs usually have very different architectures: number of layers, hidden sizes..etc; this makes applying cross-family transfer hardly possible.
> Additionally, by design LoRA works by learning a *small*, low-rank update to the original weight matrix. "*Cross-family transfer*" will almost certainly result in a degraded or nonsensical model performance. By extension, this applies to MoL adapters.
>
> **Reviewer rJ7J**: *Include downstream quantitative tasks.*
>
> The core contribution of our work is a **parameter-efficient multimodal pretraining strategy** that enables generation. We position our work alongside pretraining-focused methods like [1], which also primarily assesses performance via per-modality loss. They compare to two concurrent methods, e.g. a dense transformer (Chameleon) and a MoE-based architecture, which they train themselves. We compare MoL to a single, shared **LoRA adapter**. This is the most equivalent parameter-efficient baseline for multimodal pretraining.
>
> We acknowledge the value of downstream task evaluation. We are actively working to produce relevant benchmarks against existing fine-tuning methods. However, we note that most comparable methods [2,3,4] are typically fine-tuned *directly on* those downstream tasks (e.g., VQA, Captioning) and often rely on **Instruct-like models**, making a direct, apples-to-apples comparison difficult without extensive retraining. Our primary focus remains on **multimodal generation enabled by pretraining**.
>
> [1] Mixture-of-transformers: A sparse and scalable architecture for multi-modal foundation models. Weixin Liang, et al. in TMLR, 2025.
>
> [2] Visual instruction tuning. Haotian Liu, Chunyuan Li, Qingyang Wu, and Yong Jae Lee. NeurIPS 2023.
>
> [3] Qwen-VL: A Versatile Vision-Language Model for Understanding, Localization, Text Reading, and Beyond. Jinze Bai, et al. 2023
>
> [4] Sharegpt4v: Improving large multi-modal models with better captions. Lin Chen, et al. 2023.

---

### Official Review · Reviewer_deWW · 2025-10-31

**Soundness:** 2
**Presentation:** 2
**Contribution:** 2
**Rating:** 2
**Confidence:** 4

**Summary:**

This paper proposes a mixture of LoRA inspired by the mixture of transformers (MoT) for effective multimodal fine-tuning for understanding and generative tasks. Overall, the idea of this work is incremental and lacking quantitative results and analysis, and the paper is still in a pre-mature stage for submission (much improvement required for a more comprehensive experiment and comparison with existing works).

**Strengths:**

1) The paper investigates a mixture of LoRA inspired by the mixture of transformers (MoT) for multimodal fine-tuning, studying the potential of replacing MoT with LoRA.
2) The method is well described and straightforward to understand, with no complex theory or module required.
3) The choice experiment is justifiable; it would be good to make it more comprehensive.

**Weaknesses:**

1) The proposed method appears to be a simple replacement of LoRA for MoT? which sounds subliminal as a contribution.
2) Comparison with the baseline MoT is missing. Why?
3) The results are presented in terms of the loss curves, which could be improved to make them a comprehensive assessment. The current form could hardly reflect the actual performance in multimodal tasks.
4) It would be better to show quantitative comparison in a Table form for all experiments.
5) There is a lack of comprehensive comparative analysis to show how this work challenges SOTA; comparing with baseline LoRA is insufficient.

**Questions:**

NA

---

> ### Author Response · Authors · 2025-11-20
> **Answer to Reviewer deWW**
>
> We are grateful to **Reviewer deWW** for their comments. Allow us to provide some additional information and answer a few interrogations on some of the points raised by **Reviewer deWW**.
>
> First, we thank **Reviewer deWW** for mentioning that our method is "*well described and straightforward to understand*". Could they provide some additional guidance on how to improve clarity given their *Presentation* rating of 2/4?
>
> **Reviewer deWW** mentions how we are "*studying the potential of replacing MoT with LoRA*" and later mentions in the weakness section that MoL is "*a simple replacement of LoRA for MoT*". However, as mentioned on several occurences throughout the manuscript (e.g. lines 68-70, 294-296,) our proposed method, Mixture of LoRA (MoL) does not aim at replacing MoT as their respective purposes **are complementary**. MoT architectures requires **training a model from scratch** and necessitates to be trained on trillions of tokens, on the contrary MoL is a **fine-tuning** method that leverages pre-trained text-only LLMs as the base model and fine-tunes it to generate and understand multimodal tokens.
>
> **Reviewer deWW**: *Comparison with the baseline MoT is missing. Why?*
> The weights of the trained MoT models are unfortunately not publicly available, and despite our inquiries to the authors we have not managed to get any information on the original MoT models' metric performances. Moreover, we do not dispose of the necessary compute to retrain from scratch the models discussed in the original MoT paper. Without retraining, the only possibility for comparing our approach to MoT would be to compare loss curves. However, we believe this is not relevant for the following reasons:
> - Training data is different between MoT and MoL.
> - Overall training path cannot directly be compared as MoT trains **from scratch** while MoL finetunes a pretrained model.
>
> **Reviewer deWW**: *The results are presented in terms of the loss curves, which could be improved to make them a comprehensive assessment. The current form could hardly reflect the actual performance in multimodal tasks.*
>
> We have followed the approach proposed in [1] which focuses on the multimodal pretraining stage. In their paper, the author solely rely on the per-modality loss (as we propose to do in the present paper) to assess the performance of their method. They compare to two concurrent methods, e.g. a dense transformer (Chameleon) and a MoE-based architecture, which they train themselves. We propose comparing our method to their equivalent in the case of fine-tuning methods: a single shared LoRA adapter.
>
> **Reviewer deWW**: *It would be better to show quantitative comparison in a Table form for all experiments. There is a lack of comprehensive comparative analysis to show how this work challenges SOTA; comparing with baseline LoRA is insufficient.*
>
> We will try to produce relevant benchmarks to compare to existing fine-tuning methods. However, let us mention here that most fine-tuning models to which we could compare [2,3,4] also fine-tune the model on the evaluated downstream tasks. On the contrary our method primarely focuses on *next-token prediction*. Also, those method rely on Instruct-like model which can more easily perform on benchmarks.
>
> [1] Mixture-of-transformers: A sparse and scalable architecture for multi-modal foundation models. Weixin Liang, LILI YU, Liang Luo, Srini Iyer, Ning Dong, Chunting Zhou, Gargi Ghosh, Mike Lewis, Wen tau Yih, Luke Zettlemoyer, and Xi Victoria Lin in TMLR, 2025.
>
> [2] Visual instruction tuning. Haotian Liu, Chunyuan Li, Qingyang Wu, and Yong Jae Lee. NeurIPS 2023.
>
> [3] Qwen-VL: A Versatile Vision-Language Model for Understanding, Localization, Text Reading, and Beyond. Jinze Bai, Shuai Bai, Shusheng Yang, Shijie Wang, Sinan Tan, Peng Wang, Junyang Lin, Chang Zhou and Jingren Zhou. 2023
>
> [4] Sharegpt4v: Improving large multi-modal models with better captions. Lin Chen, Jinsong Li, Xiaoyi Dong, Pan Zhang, Conghui He, Jiaqi Wang, Feng Zhao, and Dahua Lin. 2023.

---

### Official Review · Reviewer_39vi · 2025-11-02

**Soundness:** 2
**Presentation:** 3
**Contribution:** 2
**Rating:** 4
**Confidence:** 4

**Summary:**

This paper proposes Mixture-of-LoRA (MoL), a parameter-efficient method for adapting pretrained text-only LLMs to multimodal tasks. The approach injects modality-specific LoRA adapters into frozen LLM layers while maintaining global self-attention for cross-modal fusion. The authors evaluate MoL on text-image and text-image-audio tasks using models up to 3B parameters, demonstrating improved training stability compared to vanilla LoRA baselines.

**Strengths:**

-The combination of LoRA's efficiency with MoT's modality-specific design is sensible and addresses a clear need for cost-effective multimodal adaptation.
-The paper is well-written with good visual aids that make the method easy to understand.
-The ablation studies provide valuable insights, particularly the finding that text adapters are necessary even when adapting to new modalities.
-The approach enables both understanding and generation of multimodal content, which is more general than understanding-only methods.

**Weaknesses:**

-This paper only reports pretraining losses without downstream tasks evaluation (VQA, image captioning, etc.). Loss curves alone are insufficient to demonstrate the method's practical utility.

-While MoL is more parameter-efficient than MoT, the core idea of modality-specific processing is directly borrowed. The main difference is using LoRA instead of full weight matrices, which feels incremental. A more thorough comparison with MoT (ideally empirical) would strengthen the positioning.

-The baseline LoRA setup may be suboptimal (same learning rate for all modalities).

-Section 5.2 reveals that adding MoL to FFN layers causes loss divergence despite extensive hyperparameter tuning. This suggests the method may be brittle and require careful tuning.

**Questions:**

-This paper only reports pretraining losses without downstream tasks evaluation (VQA, image captioning, etc.). Loss curves alone are insufficient to demonstrate the method's practical utility.

-While MoL is more parameter-efficient than MoT, the core idea of modality-specific processing is directly borrowed. The main difference is using LoRA instead of full weight matrices, which feels incremental. A more thorough comparison with MoT (ideally empirical) would strengthen the positioning.

-The baseline LoRA setup may be suboptimal (same learning rate for all modalities).

-Section 5.2 reveals that adding MoL to FFN layers causes loss divergence despite extensive hyperparameter tuning. This suggests the method may be brittle and require careful tuning.

-Some loss curves are hard to read due to overlapping lines.

---

> ### Author Response · Authors · 2025-11-20
> **Answer to Reviewer 39vi**
>
> We sincerely thank **Reviewer 39vi** for their insightful feedback, which has helped us to clarify and strengthen our submission.
>
> **Reviewer 39vi**: *This paper only reports pretraining losses (...)*
>
> The core contribution of our work is a **parameter-efficient multimodal pretraining strategy** that enables generation. We position our work alongside pretraining-focused methods like [1], which also primarily assesses performance via per-modality loss. They compare to two concurrent methods, e.g. a dense transformer (Chameleon) and a MoE-based architecture, which they train themselves. We compare MoL to a single, shared LoRA adapter. This is the most equivalent parameter-efficient baseline for multimodal pretraining.
>
> We acknowledge the value of downstream task evaluation. We are actively working to produce relevant benchmarks against existing fine-tuning methods. However, we note that most comparable methods [2,3,4] are typically fine-tuned *directly on* those downstream tasks (e.g., VQA, Captioning) and often rely on **Instruct-like models**, making a direct, apples-to-apples comparison difficult without extensive retraining. Our primary focus remains on **multimodal generation enabled by pretraining**.
>
> **Reviewer 39vi**: *(...) a more thorough comparison with MoT would strengthen the positioning.*
>
> We acknowledge the reviewer's perspective regarding the perceived incrementality of our approach. However, we respectfully suggest that innovation in highly developed fields often lies in the strategic refinement and combination of existing concepts. One might similarly characterize Variational Autoencoders (VAEs) as an incremental modification of standard Autoencoders by enforcing a Gaussian prior; yet, this specific constraint enabled entirely new applications in generative modeling.
>
> We believe our contribution represents a significant step forward as MoL enable fine-tuning pretrained language models to both understand and **generate** multimodal tokens. This design allows for significant savings in compute costs by effectively leveraging the extensive knowledge contained within the pretrained base model.
> Moreover, to our knowledge, MoL is the first fine-tuning method for LLM that enables multimodal generation.
>
> We agree that an empirical comparison to MoT would be ideal. Unfortunately, the trained MoT model weights are not publicly available, and we could not obtain the necessary performance metrics from the authors. Critically, we do not possess the compute resources to retrain the MoT models discussed in their original paper from scratch. This makes a direct empirical comparison impossible. We have therefore focused on comparing against the most robust and publicly available parameter-efficient fine-tuning baseline (LoRA).
>
> **Reviewer 39vi**: *The baseline LoRA setup may be suboptimal (same learning rate for all modalities).*
>
> We believe there may be a misunderstanding regarding the standard LoRA baseline.
>
> In the standard LoRA setup (the baseline we use), a **single, shared LoRA adapter** is applied across all modalities. Since there is only one set of LoRA parameters, only **one single learning rate** can be assigned to these parameters. There is no mechanism to assign different learning rates per modality. This limitation is precisely the problem our **Mixture-of-LoRA (MoL)** approach is designed to solve. By allowing for distinct LoRA adapters for each modality, MoL inherently allows the necessary latitude to set **modality-specific training hyperparameters**, including different learning rates.
>
> **Reviewer 39vi**: *Section 5.2 reveals (...) method may be brittle and require careful tuning.*
>
> Our most recent experiments in the **three modality setting** actually show how adding the MoL adapters to the FFN actually enabled quicker and more stable learning countrary to the Chameleon setting.  We posit that as complexity increases, the attention-only adapters cannot fully model the intra-modality transformations required by all modalities. The increased complexity of the three-modality task likely acts as a *natural regularizer* allowing the model to avoid divergence as observed in Fig. 3. (a) of the manuscript.
> We will discuss those findings in a new version of the manuscript to be posted before the end of discussion.
>
> Additionally, let us also mention here that even unimodal standard LoRA fine-tuning often requires careful tuning of the hyperparameters.
>
> [1] Mixture-of-transformers: A sparse and scalable architecture for multi-modal foundation models. Weixin Liang, et al. in TMLR, 2025.
>
> [2] Visual instruction tuning. Haotian Liu, Chunyuan Li, Qingyang Wu, and Yong Jae Lee. NeurIPS 2023.
>
> [3] Qwen-VL: A Versatile Vision-Language Model for Understanding, Localization, Text Reading, and Beyond. Jinze Bai, et al. 2023
>
> [4] Sharegpt4v: Improving large multi-modal models with better captions. Lin Chen, et al. 2023.

---

### Meta-Review · Area_Chair_yxeS · 2026-01-07

**Summary:**

Based on the provided reviews, the key concerns that informed the suggested decision can be summarized into four main categories: 1）Lack of Downstream Task Evaluation and Insufficient Evidence for Practical Utility；2) Limited Perceived Novelty and Incomplete Comparative Analysis; 3) Flaws in Experimental Setup and Result Analysis; 4) Failure to Probe Critical Issues and Failure Cases.

**Reviewer Concerns:**

The reviewers collectively find that the paper attempts to validate a method with limited perceived novelty based on insufficient evidence (no downstream tasks) and with notable flaws in experimental design and analysis. Therefore, unless the authors can provide extensive supplementary experiments (most crucially, comprehensive downstream task evaluation) along with more rigorous analysis and comparisons, the paper in its current state fails to convincingly demonstrate the significance and value of its contribution.

**Reviewer Scores:**

The reviewer's most critical concern was not resolved and is a significant barrier to a higher score. The reviewer would likely keep their original score, reflecting that some concerns were alleviated, but the fundamental issues remains outstanding. The paper is seen as having a clearer, more defensible contribution but still lacking the conclusive evidence required for acceptance.

---

### Decision · Program_Chairs · 2026-01-26

Reject